# Anti-Diabetic Activity of Polysaccharides from *Auricularia cornea* var. Li.

**DOI:** 10.3390/foods11101464

**Published:** 2022-05-18

**Authors:** Yuan Fu, Liwen Wang, Guochuan Jiang, Lili Ren, Liyan Wang, Xuejun Liu

**Affiliations:** 1College of Food Science and Engineering, Jilin Agricultural University, 2888 Xincheng Street, Changchun 130118, China; 20201617@mails.jlau.edu.cn (Y.F.); jiangguochuan@jlau.edu.cn (G.J.); wangliyan@jlau.edu.cn (L.W.); 2China South Korea (Changchun) International Cooperation Demonstration Zone Management Committee, Changchun 130042, China; wlw77458350@163.com; 3Key Laboratory of Bionic Engineering (Ministry of Education), College of Biological and Agricultural Engineering, Jilin University, 5988 Renmin Street, Changchun 130022, China; liliren@jlu.edu.cn

**Keywords:** *Auricularia cornea* var. Li., polysaccharides, diabetes mellitus II

## Abstract

*Auricularia cornea* var. Li. polysaccharide (ACP) has many important biological activities and has potential application value in food engineering, pharmaceutical science, and health care. The results were as follows: the extraction rate of ACP was 28.18% ± 1.41% and the purity of ACP was 86.92% ± 2.80%. ACP contains mannitol 32.41%, glucuronic acid 6.96%, rhamnose 0.32%, glucose 42.35%, galactose 0.77%, xylose 16.83%, and fucose 0.36%, without galacturonic acid and arabinose. In addition, the results of an animal test of diabetes mellitus II (DM II) with ACP showed that the total cholesterol (TC), triglyceride (TG), low-density lipoprotein cholesterol (LDL-C), and fasting blood glucose and water in the serum of mice with ACP were significantly lower than those in the model group; the serum SOD, hepatic glycogen, and insulin of mice added with ACP were significantly higher than those in the model group. More importantly, ACP had no significant adverse effects on organ index and liver and kidney tissue morphology in mice. These results suggest that ACP can be used as a potential functional food component for the prevention or treatment of diabetes.

## 1. Introduction

Diabetes is a common, multi-etiological, and hereditary disorder of endocrine and metabolic disorders [1], which has become the third major threat to human health today. The number of people with diabetes mellitus II (DM II) accounts for about 90% of all types of diabetes [2]. At present, the clinical treatment of type II diabetes is mainly injecting insulin, oral sulfonylurea, and BIS guanidine as hypoglycemic drugs so as to achieve the goal of lowering blood sugar. These drugs have strong and quick effects, but they all have different degrees of toxicity and side effects [1]. Therefore, it is necessary to find new and safe hypoglycemic active ingredients, which have become a hot spot in the fields of biology and medicine.

Edible fungi represent the general name of large fungi that can be eaten or used as medicine [3]. Edible fungus polysaccharide has high research value and has biological activities such as antioxidant [4,5], hypoglycemic [6,7,8], anti-inflammatory [3,9], hypolipidemic [10], liver protection [11], antitumor (for example, a polysaccharide isolated from *Calocybe indica* var. APK2 also showed effective antitumor activities [12]. The cytotoxic effect of this polysaccharide was evaluated in vitro on HeLa cells (Human cervical cancer) and normal cells NIH3T3 (Murine fifibroblat) by the MTT assay. A water-soluble polysaccharide (TOP-2, Mw: 63 kDa) was isolated from the basidiomycetous fungus *Trametes orientalis* [13]. TOP-2 showed antitumor activities in Lewis lung carcinoma (LLC) tumor-bearing mice. Previous studies have shown that the polysaccharide aamp-n extracted from edible fungi can significantly reduce the blood glucose level and improve lipid metabolism in diabetic mice. In addition, islets are also protected from damage [14]. In a randomized double-blind trial, supplementation of *Agaricus Blazei Murill* extract combined with gliclazide and metformin for 12 weeks may reduce insulin resistance (IR) in patients with type 2 diabetes mellitus (DM II) [14]. Edible fungi can significantly reduce blood lipid levels, including total cholesterol (TC), triglycerides (TG), low-density lipoprotein cholesterol (LDL-C), and high-density lipoprotein cholesterol (HDL-C). Edible fungi also contain a lot of fiber and water. They are rich in natural insulin-like enzymes that help break down glucose in food and reduce IR. They also contain different compounds that promote the good function of the pancreas and liver, thus contributing to the production and release of insulin, thereby confirming healthy metabolic function. Most medicinal edible fungi, such as *Ganoderma lucidum*, *Inonotus obliquus*, *Pleurotus* spp., *Phellinus linteus*, *Poria cocos,* and *Sparassis crispa,* have shown beneficial effects in the treatment of DM [15].

*Auricularia cornea* var. Li., as a common edible fungus, has a wide variety. In China, cultivated edible *Auricularia auricula* is divided into the following two kinds: *Auricularia cornea* and *Auricularia*
*h**eimer* [16]. *Auricularia cornea* var. Li. was selected by the team of Jilin Agricultural University and domesticated by the team of Academician Li Yu of Jilin Agricultural University into a stable genetic natural mutant pure white strain of *Auricularia cornea*. It is a new edible fungus variety with a high yield and high quality. *Auricularia cornea* var. Li. polysaccharide, the main active substance of *Auricularia cornea* var. Li., has a variety of biological activities, such as antioxidant [15], anti-fatigue [15], anti-inflammatory [17], and liver-protective activities [18]. Its anti-diabetic activity has not been reported yet. This experiment will study the anti-diabetic activity of *Auricularia cornea* var. Li. polysaccharide.

## 2. Materials and Methods

### 2.1. Materials and Reagents

*Auricularia cornea* var. Li. obtained from Jilin Agricultural University, Jilin Province. The following chemicals were used: The commercial kits of Total cholesterol (T-CHO) kit, Triglyceride (TG) kit, High-density lipoprotein cholesterol (HDL-C), Low-density lipoprotein cholesterol (LDL-C), Superoxide dismutase (SOD), Ins ELISA Kit (INS) and Glycogen test box were obtained from Nanjing Biotechnology Co. Ltd. (Nanjing, China). Hgm-114 blood glucose meter Omron Automation Co., Ltd. AS1 blood glucose test paper Omron Automation Co., Ltd. All reagents used were analytical grade.

### 2.2. Extraction of ACP

ACP was extracted using the method of [19], The proteins in crude ACP were removed using the Sevag method [20]. Yield [21], extraction rate [22], and purity [23] of ACP.

### 2.3. Monosaccharide Compositions of ACP

Monosaccharide compositions of ACP were estimated through Gas chromatography–Mass Spectrometry (GC–MS) using 6890-5975 gas chromatography–mass spectrometer (Thermo Fisher Scientific Co., Ltd., Shanghai, China) [24].

### 2.4. Animal Experimental Design

Fifty male Kunming mice (8-week-old, 20 ± 2 g) (Liaoning Changsheng Biotechnology Co., Ltd., Benxi, China) were fed a basal diet. Afterward, all animals were raised under the following conditions: temperature of 24 °C and light/dark cycle of 12 h/12 h. They were allowed to drink water and eat food freely. Ten mice were chosen at random as a normal control group and fed the basal diet (3.40 Kcal/g, protein 23.07%, fat 11.85%, and carbohydrate 65.08%, Beijing Keao Xieli Feed Co., Ltd.). The remaining mice were fed a high-fat and sugar diet (4.7 Kcal/g, protein 19.80%, fat 35.20%, and carbohydrate 45.00%, Beijing Botai Hongda Biotechnology Co., Ltd., batch No.: HD001) for 30 consecutive days. During this period, mice were weighed regularly to observe changes in body weight and food intake. All trials were carried out according to the Jilin Agricultural University laboratory guidelines for animals. The protocol was authorized by the Jilin Agricultural University Institutional Animal Care and Use Committee. The animal protection directive aims to protect animals in scientific research. Furthermore, it announces the implementation of the 3Rs principle (replacement, reduction, and refinement). Animal model of this experiment is in line with the 3Rs policy.

After successful establishment of the diabetes model, mice were randomly divided into the following 5 groups: normal control group, diabetes group, positive control group (metformin hydrochloride, 100 mg/kg body weight), ACP low-dose group (100 mg/kg body weight), and ACP high-dose group (400 mg/kg body weight) [25]. Mice were given regular gavage every day and drank freely. The mice should be weighed, food intake and recorded before gavage every day, and then gavage for 30 days. Fasting blood glucose should be measured every 7 days. On the last day of treatment (day 31), drinking water and food were no longer provided to all mice. After 12 h of food deprivation, the weight of mice was weighed, and the blood glucose was measured and recorded.

### 2.5. Determination of Visceral Indexes

After the experiment, the heart, spleen, liver, kidney, thymus, and other organs were dissected. The viscera were weighed and recorded to analyze the visceral index. The calculation formula is as follows [24]:(1)Visceral index (g/100 g)=M1/M ×100% 

M1: organ mass, M: mouse body weight.

### 2.6. Biochemical Analysis

After the last administration, the mice were weighed, and their body weight was recorded after 12 h of fasting. Blood was collected from eyeballs, centrifuged (3000× *g*, 10 min) to obtain serum, and stored in refrigerator at 4 ℃. The contents of TC, TG, LDL-C, HDL-C, SOD, liver glycogen, and INS in serum were determined by commercial kit method.

### 2.7. Preparation of Tissue Hematoxylin Eosin (HE) Stained Sections

The dissected liver and kidney tissues of mice were immediately soaked in 10% neutral formalin solution for fixation. After changing the fixation solution three times, the organ tissues to be observed were taken out from the fixation solution. Firstly, gradient ethanol dehydration (70%, 80%, 95%, 100%) was carried out. Xylene was used for transparent treatment in a ventilated environment, and then embedded with an embedding machine (KH-BL, Hubei Xiaogan kuohai Medical Technology Co., Ltd., Xiaogan, China) to make paraffin sections (RM2125RT paraffin slicer, Leica, Germany), The sliced tissue was stained with he, washed off the excess dye, sealed with neutral gum, and then observed under the microscope (BX53 microscope, Olympus, Japan).

### 2.8. Statistical Analysis

All data were expressed as the mean ± standard deviation (SD) from 6 independent experiments and compared using one-way analysis of variance (ANOVA) by employing SPSS 23.0 software and Dunnett’s test. In addition, *p*-value < 0.05 was considered significant.

## 3. Results

### 3.1. Extraction Rate, Purity, and Monosaccharide Composition of ACP

The extraction rate and purity of ACP were 28.18% ± 1.41% and 86.92% ± 2.80%. The monosaccharides of ACP were 32.41% mannitol, 6.96% glucuronic acid, 0.32% rhamnose, 42.35% glucose, 0.77% galactose, 16.83% xylose, and 0.36% fucose, without galacturonic acid and arabinose. This result is similar to that of Wang et al. [19] on the monosaccharide composition of polysaccharides isolated from *Auricularia cornea* var. Li.

### 3.2. Body Weight and Feed Intake

#### 3.2.1. Effects of ACP on Body Weight in DM II Mice

The effects of different doses of ACP on the bodyweight of DM II mice are shown in Table 1. It can be seen from Table 1 that at the beginning of the test, the initial weight of mice in each group did not change significantly, and their weight increased with the increase in test time. By comparing the bodyweight after modeling in the table, it can be seen that the bodyweight of mice in the normal group is higher than that of mice in the other groups (*p* < 0.01). After four weeks of treatment, the bodyweight of mice in each dose group and positive control group was significantly lower than that of mice in the model group (*p* < 0.01 or *p* < 0.05). There was no significant difference in bodyweight between the high-dose agaric polysaccharide group and the normal group and the positive control group (*p* > 0.05). The results showed that ACP significantly inhibited the weight gain of mice induced by type II diabetes.

#### 3.2.2. Effects of ACP on Feed Intake in DM II Mice

Overeating is considered to be one of the typical symptoms caused by diabetes. According to the food intake of mice shown in Table 2, the addition of ACP showed beneficial effects on diabetes mice. There was a significant difference in food intake between ACP high dose and the model (*p* < 0.01). The food intake levels of the ACP low dose and model were similar. Compared with model mice with poor condition, positive control and ACP high dose performed well, and diabetes index improved.

### 3.3. Blood Analysis

#### 3.3.1. Effect of ACP on Blood Glucose (GLU) in Mice

As shown in Table 3, the blood glucose level of DM II mice after feeding with ACP had an obvious downward trend. After feeding the special high-fat and high-sugar diet to mice, the fasting blood glucose of mice increased exponentially (STZ < 0.01) after the injection of the DM II drug (*p* < 0.01), and the blood glucose value did not fall for a few days, indicating that the model was successful. Compared with the blood glucose values of mice in the model group, the blood glucose level of mice treated with each dose of ACP decreased significantly (*p* < 0.01). It can be seen from the numerical value that the hypoglycemic effect of the high-dose ACP group is more obvious than that of the low-dose group, indicating that the anti-diabetic activity of ACP is dose-dependent. However, there was still a difference in blood glucose compared with the normal group (*p* < 0.01). The data analysis shows that ACP can improve the fasting blood glucose of diseased mice.

#### 3.3.2. Effects of ACP on Blood Lipid Related Indexes in DM II

The effects of ACP on TC and TG content in the serum of diabetic mice are shown in Figure 1. As shown in Figure 1, compared with the normal group mice, the levels of TC and TG in the model group mice increased significantly (*p* < 0.01), showing a serious disorder of lipid metabolism. From the ACP treatment group, it can be seen that the high-dose ACP group can significantly inhibit the increase in the serum TC and TG levels in diabetic mice (*p* < 0.01), while the low-dose ACP group has no significant effect, but it also has a certain curative effect. There was no significant difference in TC and TG between the high-dose ACP group and the normal group. The effect of ACP on the levels of HDL-C and LDL-C in the serum of diseased mice is shown in Figure 1. It can be seen from Figure 1 that the function of lipid metabolism in diabetic mice has been disturbed. Compared with normal mice, the HDL-C value of mice in the model group has obviously decreased (*p* < 0.01), while the LDL-C value has increased significantly (*p* < 0.01). However, the treatment of ACP can reduce the risk of cardiovascular disease by regulating dyslipidemia in diabetic mice. The HDL-C level and LDL-C level of the low-dose ACP group and high-dose ACP group are not significantly different from those of the normal group. However, it can be seen from the data that the HDL-C level and LDL-C level of the high-dose ACP group are closer to the normal group, indicating that the activity of ACP alleviates the lipid metabolism disorder of diabetic mice increases with an increase in dose.

#### 3.3.3. Effects of ACP on SOD in DM II Mice

According to Table 4, compared with normal mice, the level of serum SOD in the model group decreased significantly (*p* < 0.01), indicating that diabetic mice were injured by oxidative stress. After being given different doses of ACP, the activity of SOD in the serum of diabetic mice increased significantly (*p* < 0.01). Among them, the SOD activity of mice in the high-dose ACP group was higher than that in the metformin positive control group, and there was no significant difference between the high-dose ACP group and the normal group (*p* > 0.05). In addition, the SOD activity of mice in the positive control group treated with metformin was also significantly higher than that in the model group (*p* < 0.01).

#### 3.3.4. Effect of ACP on Liver Glycogen Level in Mice

The effect of ACP on liver glycogen in diabetic mice is shown in Table 4. From the table, it is obvious that the level of liver glycogen in the model group is significantly lower than that in the normal group (*p* < 0.01). After four weeks of treatment with high-dose ACP, the recovery effect of mouse liver glycogen value was significantly better than that of the model group (*p* < 0.01), but there was no significant difference between the low-dose group and the model group (*p* > 0.05), indicating that the treatment effect increased in a dose-dependent manner. The results showed that ACP showed hypoglycemic activity by regulating glycogen production.

#### 3.3.5. Effect of ACP on Insulin Level in Mice

As shown in Table 4, due to the destruction of STZ β-Cells, resulted in a decrease in the ratio of the amount of insulin secreted by mice to the normal amount. The values of serum insulin and pancreatic insulin in the model group were significantly different from those in the normal group (*p* < 0.01). After treatment with different doses of ACP and drugs for one month, compared with the model group, the levels of serum insulin and pancreatic insulin increased significantly (*p* < 0.01), and the treatment effect was positively correlated with the dosage. By comparing the high-dose ACP group with the model group, it can be seen that ACP may reduce blood glucose by repairing islet cells, promoting insulin secretion, improving carbohydrate metabolism by increasing insulin level and sensitivity, and promoting liver glycogen synthesis.

### 3.4. Body Tissues

#### 3.4.1. Analysis of Organ Index

The effects of ACP on the liver, kidney, heart, pancreas, and spleen of DM II mice are shown in Table 5. There was no significant difference in heart index between each group (*p* > 0.05), indicating that DM II had no significant effect on the hearts of mice at the end of the trial.

As shown in Table 5, the liver index of mice in the model group was significantly higher (*p* < 0.01) than that of mice in the normal group. This result showed that a high-fat and high-sugar diet combined with STZ injection caused continuous hyperglycemia in mice so that the liver could not maintain the dynamic balance of glycogen normally, causing certain damage to the liver of mice, and its symptom was liver hypertrophy. Under the intervention of low-dose and high-dose ACP, the liver index of mice was significantly lower than that of the model group (*p* < 0.01), indicating that ACP has a significant protective effect on the liver of diseased mice. There was no significant change in the liver index of mice in the high-dose ACP group and the positive control group compared with the normal group (*p* > 0.01).

It can be seen from Table 5 that the value of the kidney index in the model group is higher than that in the normal group (*p* < 0.01), which shows that it has an effect on the kidneys of diseased mice. Under the intervention of low-dose and high-dose ACP compared with the model group, the value decreased significantly (*p* < 0.01), indicating that ACP has a protective effect on the kidneys of diseased mice. There was no significant difference in renal index between mice treated with high-dose ACP and the normal group (*p* > 0.01).

As shown in Table 5, there is a great difference between the pancreatic index of the model group and the normal group (*p* < 0.01), indicating that their pancreas has been damaged to some extent. The pancreatic index of mice in the low-dose and high-dose groups was significantly reduced (*p* < 0.01 or *p* < 0.05), and the pancreatic index of the high-dose group had no significant difference compared with the normal group (*p* > 0.05), indicating that ACP had certain protective and repair effects on the pancreas of diabetic mice.

#### 3.4.2. Effects of ACP on Organs of DM II Mice

(1)Liver

Liver biopsy analysis is shown in Figure 2. It can be seen from the figure that the hepatocytes in the normal group are arranged orderly, the nucleus is complete, the appearance is spherical, and the boundary between cells is clearly visible. In the model group, there was no obvious boundary between cells, the nucleus was incomplete, the lipid droplets in the cytoplasm were serious, there were vacuoles, and the inflammatory cells infiltrated a large area, indicating that the liver was damaged. The ACP group and the positive control group had a certain degree of improvement compared with the model group. The vacuoles in liver cells were reduced and arranged more orderly, and liver injury was reduced. There was a gap between the low dose of the ACP group and the other two groups, but it could be explained that ACP had a protective effect on the liver of diabetic mice.

(2)Kidney

The Nephridium biopsy analysis is shown in Figure 3. The overall structure of the glomerulus in the normal group is complete. In the model group, renal pathological damage such as glomerulosclerosis, cellular inflammatory infiltration, and severe edema occurred. There was no difference between the positive control group and the normal control group. In the high-dose agaric ACP group, the kidney almost returned to normal. These results indicate that ACP has the most significant effect on protecting the kidney from injury in diabetic mice.

## 4. Discussion

*Auricularia cornea* var. Li. is a rare edible fungus with food and drug homology recently, but there are few studies on its activity, and its anti-diabetes activity has not been reported. In this paper, agaric polysaccharide was extracted by experiment, and its anti-diabetic activity was studied by a mouse experiment, which can be used as a reference for other activity studies in the future.

The monosaccharide composition of ACP extracted in this experiment mainly consists of mannose (32.41%), glucose (42.35%), and xylose (16.83). This result is similar to the monosaccharide composition of polysaccharides isolated from *Auricularia cornea* var. Li. by Wang et al. [19], in which the contents of glucose, mannose, and xylose are significant, but the proportion is different. In addition, the monosaccharide composition of polysaccharides in *Auricularia auricula* was also studied. The isolated neutral polysaccharides were mainly glucose, the acidic polysaccharides were mainly mannose and glucuronic acid, and the other monosaccharides were heteropolysaccharides. The reason may be that they are edible basidiomycetes and fungi, all of which belong to the family *auriculariae* [19].

DM II patients usually show symptoms such as hyperglycemia, excessive drinking of water, overeating, and weight loss, and are usually accompanied by dyslipidemia, such as increased levels of TC, TG, and LDL and decreased content of HDL [26]. In our experiment, compared with the normal group (Figure 1), these characteristics were obviously shown in the model group. The increase in food intake in DM II mice may be due to the impairment of energy metabolism caused by chronic hyperglycemia [27]. Weight loss and fasting blood glucose can be explained by the excessive consumption of muscle and protein tissue caused by impaired blood glucose metabolism in DM II mice [27]. After ACP administration, the food intake (Table 2), body weight (Table 1), FBG (Table 3), and blood lipids (Figure 1) of DM II mice were improved. We can say that ACP can improve the symptoms of DM II.

More and more evidences show that the increase in oxidative stress and the dysfunction of the antioxidant system contribute to the development, progress, and pathogenesis of diabetes and its complications in clinical and experimental aspects [28]. It is necessary to explore the clinical treatment of antioxidant stress as an additional standard treatment for patients with diabetes and its complications [28]. Yao et al. [29] proved that SOD can catalyze the conversion of superoxide radicals into hydrogen peroxide, which is then decomposed into water and oxygen by GSH PX and cat, preventing the formation of ROS. The antioxidant activity of the enzyme may be a suitable indirect marker for evaluating the oxidant–antioxidant status in tissue [30]. In our experiment, compared with the normal group (Table 4), the level of serum SOD in the model group was significantly lower (*p* < 0.01), indicating that DM II mice were damaged by oxidative stress. After administration of different doses of ACP, the serum SOD activity of DM II mice increased significantly (*p* < 0.01). Among them, the SOD activity of the high-dose ACP group was higher than that of the positive group, and there was no significant difference compared with the normal group (*p* > 0.05). Therefore, ACP treatment can increase SOD activity and further reduce free radicals in the DM II body so as to enhance the antioxidant capacity of the body.

The synthesis and decomposition of liver glycogen are regulated by insulin and glucagon. The content of glycogen can express the body’s ability to convert glucose into glycogen, which proves the body’s response efficiency to insulin. In our experiment, the level of liver glycogen in the model group (Table 4) was significantly lower than that in the normal group (*p* < 0.01). After four weeks of high-dose ACP treatment, the recovery effect of liver glycogen value in mice was significantly better than that in the model group (*p* < 0.01). Therefore, ACP can increase the content of liver glycogen, enhance the synthesis of liver glycogen, reduce the decomposition of glycogen, and reduce the output of liver glucose, which is also one of the reasons for the hypoglycemic effect.

Insulin can reduce blood sugar by promoting the uptake and utilization of glucose by body tissues and cells and inhibiting glycogen decomposition and gluconeogenesis [31]. Nie et al. [32] pointed out that restoring insulin homeostasis and reducing fasting insulin levels are very important to controlling hyperglycemia. In our experiment, after one month of ACP and drug treatment with different doses, compared with the model group (Table 4), the levels of serum insulin and pancreatic insulin increased significantly (*p* < 0.01), and the treatment effect was positively correlated with the dose. These results show that ACP treatment can improve insulin sensitivity and alleviate insulin resistance.

As sensitive organs, the liver and kidney have great detoxification ability of toxic substances and play a key role in glucose metabolism [33]. In this study, compared with the model group (Figure 2), the ACP group and the positive control group improved to a certain extent, and the liver injury was reduced. The difference between the low-dose ACP group and the other two groups can explain the protective effect of ACP on the liver of diabetic mice. It was observed in Figure 3 of the HE section that the model group had renal pathological damage, and there was no difference between the positive control group and the normal control group. In the high-dose ACP group, the kidney almost returned to normal. These results indicate that ACP has a significant role in protecting the liver and kidney from damage in diabetic mice.

## 5. Conclusions

In this study, a polysaccharide (ACP) was extracted from *Auricularia cornea* var. Li. ACP was mainly composed of mannose (32.41%), glucose (42.35%), and xylose (16.83). In the animal experiment on DM II, ACP can significantly improve the hyperglycemia symptoms of DM II mice, reduce blood lipid, reduce liver and kidney injury, and increase the contents of SOD, liver glycogen, and insulin. In general, ACP, as an antioxidant and anti-diabetic drug, has important application potential in the functional food and pharmaceutical industries. The hypoglycemic mechanism of ACP needs to be further explored in the future.

## Figures and Tables

**Figure 1 foods-11-01464-f001:**
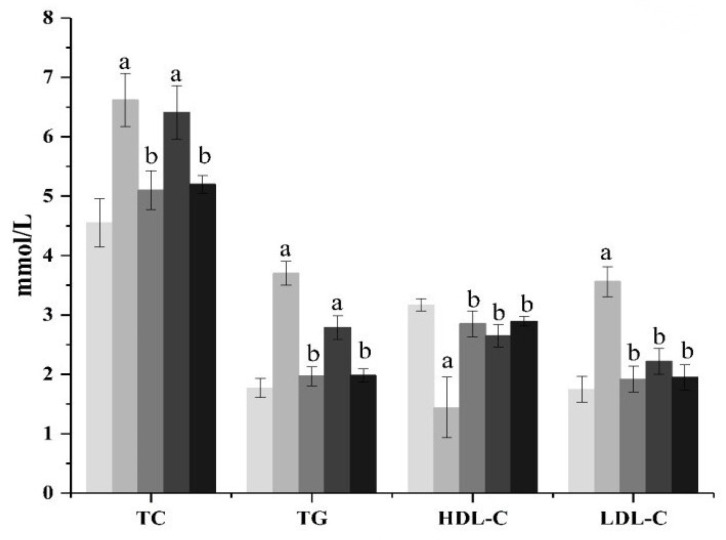
Effect of ACP on serum lipids indexes in mice. Mean values ± SD (n = 6.) a: Compared with the normal group, *p* < 0.01; b: Compared with the model group, *p* < 0.01. 1: normal, 2: model, 3: positive control, 4: ACP low dose, 5: ACP high dose.

**Figure 2 foods-11-01464-f002:**
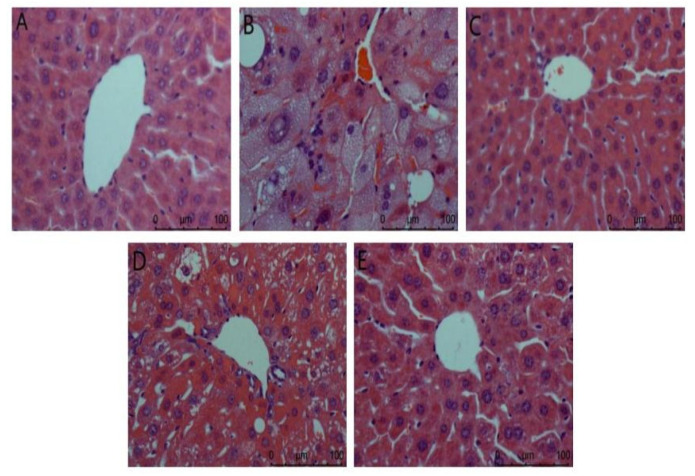
Liver biopsy analysis in mice. (**A**): Normal; (**B**): Model; (**C**): Positive control; (**D**): ACP low dose; (**E**): ACP high dose.

**Figure 3 foods-11-01464-f003:**
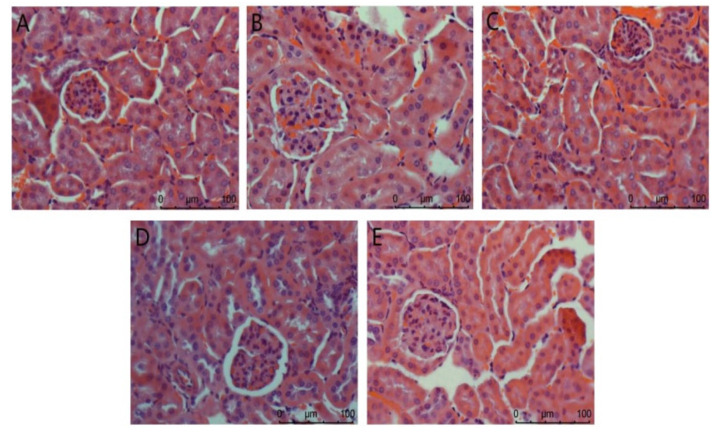
Nephridium biopsy analysis in mice. (**A**): Normal; (**B**): Model; (**C**): Positive control; (**D**): ACP low dose; (**E**): ACP high dose.

**Table 1 foods-11-01464-t001:** Effects of ACP on weight in mice.

Samples	Normal (g)	Model (g)	Positive Control (g)	ACP Low Dose(g)	ACP High Dose (g)
Initial weight	35.32 ± 1.06	35.29 ± 1.63	34.38 ± 1.31	34.18 ± 1.84	35.22 ± 2.03
Weight after modeling	37.78 ± 0.26	43.36 ± 1.46 ^a^	42.98 ± 1.23 ^a^	41.38 ± 2.52 ^a^	42.18 ± 2.78 ^a^
One week of treatment	39.86 ± 0.95	45.27 ± 1.09 ^a^	43.39 ± 1.59	43.45 ± 2.51	43.68 ± 2.39
Two weeks of treatment	42.21 ± 0.25	47.51 ± 0.86 ^a^	43.47 ± 1.22 ^b^	45.47 ± 2.40 ^c^	44.18 ± 2.15 ^b^
Three weeks of treatment	43.97 ± 0.81	49.25 ± 0.60 ^a^	43.84 ± 1.36 ^b^	47.36 ± 1.81 ^ac^	44.47 ± 1.45 ^b^
Treatment of the final	44.44 ± 0.60	50.43 ± 0.73 ^a^	44.27 ± 1.07 ^b^	48.66 ± 1.44 ^ac^	44.98 ± 1.29 ^b^

Mean values ± SD (n = 6.) ^a^: Compared with the normal group, *p* < 0.01; ^b^: Compared with the model group, *p* < 0.01; ^c^: compared with the model group, *p* < 0.05.

**Table 2 foods-11-01464-t002:** Effects of ACP on food intake in mice.

Samples	Normal (g)	Model (g)	Positive Control (g)	ACP Low Dose (g)	ACP High Dose (g)
Food intake (g/pcs/d)	5.10 ± 0.15	7.70 ± 0.18 ^a^	5.80 ± 0.13 ^b^	7.12 ± 0.15	6.50 ± 0.12 ^b^

^a^: Compared with the normal group, *p* < 0.01; ^b^: Compared with the model group, *p* < 0.01.

**Table 3 foods-11-01464-t003:** Effects of ACP on GLU in mice.

Samples	After Modeling mmol/L	One Week of Treatment mmol/L	Two Week of Treatment mmol/L	Three Week of Treatment mmol/L	Treatment of the Final mmol/L
Normal	4.67 ± 0.97	5.06 ± 0.64	5.00 ± 0.75	4.80 ± 0.86	4.70 ± 0.55
Model	16.93 ± 2.78 ^a^	18.31 ± 2.76 ^a^	19.00 ± 1.72 ^a^	20.33 ± 1.93 ^a^	20.66 ± 1.96 ^a^
Positive control	18.18 ± 4.18 ^a^	16.17 ± 3.57 ^a^	13.57 ± 2.27 ^ab^	11.20 ± 1.11 ^ab^	9.50 ± 1.33 ^ab^
Polysaccharide low dose	18.32 ± 1.71 ^a^	18.93 ± 1.59 ^a^	17.95 ± 1.29 ^a^	17.10 ± 1.05 ^ab^	16.61 ± 1.43 ^ab^
Polysaccharide high dose	16.24 ± 4.82 ^a^	17.62 ± 2.21 ^a^	14.88 ± 2.16 ^ab^	12.72 ± 2.34 ^ab^	10.42 ± 2.64 ^ab^

Mean values ± SD (n = 6.) ^a^: Compared with the normal group, *p* < 0.01; ^b^: Compared with the model group, *p* < 0.01.

**Table 4 foods-11-01464-t004:** Effects of ACP on SOD, liver glycogen, and insulin in mice.

Samples	SOD (U/mg Prot)	Liver Glycogen (mg/g)	Serum Insulin (mIU/L)	Pancreatic Insulin (mIU/L)
Normal	124.75 ± 1.07	33.12 ± 2.82	6.64 ± 0.15	24.44 ± 0.07
Model	75.52 ± 2.78 ^a^	23.00 ± 2.28 ^a^	4.28 ± 0.15 ^a^	16.89 ± 0.34 ^a^
Positive control	109.9 ± 4.18 ^ab^	28.8 0 ± 0.99 ^ab^	5.22 ± 0.16 ^ab^	22.05 ± 0.87 ^ab^
ACP low dose	106.14 ± 2.75 ^ab^	24.10 ± 1.78 ^a^	4.44 ± 0.11 ^ab^	18.60 ± 1.08 ^ab^
ACP high dose	118.29 ± 2.75 ^b^	32.01 ± 1.67 ^b^	4.88 ± 0.13 ^ab^	23.58 ± 0.81 ^b^

Mean values ± SD (n = 6.) ^a^: Compared with the normal group, *p* < 0.01; ^b^: Compared with the model group, *p* < 0.01.

**Table 5 foods-11-01464-t005:** Effects of ACP on the organ index in mice.

Samples	Heart Coefficient (g/100 g)	Liver Coefficient (g/100 g)	Kidney Coefficient (g/100 g)	Pancreas Coefficient (g/100 g)	Spleen Coefficient (g/100 g)
Normal	0.43 ± 0.04	3.96 ± 0.04	1.35 ± 0.05	0.79 ± 0.02	0.44 ± 0.02
Model	0.39 ± 0.03	7.50 ± 0.25 ^a^	2.00 ± 0.15 ^a^	0.97 ± 0.09 ^a^	0.22 ± 0.03 ^a^
Positive control	0.38 ± 0.07	4.48 ± 0.25 ^b^	1.55 ± 0.10 ^b^	0.84 ± 0.12 ^b^	0.38 ± 0.02 ^ab^
ACP low dose	0.43 ± 0.05	5.39 ± 0.51 ^ab^	1.61 ± 0.11 ^ab^	0.85 ± 0.04 ^c^	0.35 ± 0.03 ^ab^
ACP high dose	0.37 ± 0.03	4.44 ± 0.22 ^b^	1.38 ± 0.10 ^b^	0.76 ± 0.04 ^b^	0.41 ± 0.01 ^b^

Mean values ± SD (n = 6.) ^a^: Compared with the normal group, *p* < 0.01; ^b^: Compared with the model group, *p* < 0.01; ^c^: compared with the model group, *p* < 0.05.

## Data Availability

Data are contained within the current manuscript.

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
