# Peer review of "Anti-Diabetic Activity of Polysaccharides from Auricularia cornea var. Li."

_foods, 2022, doi:10.3390/foods11101464_

Round 1

Reviewer 1 Report

Introduction:

The last paragraph in the introduction should be deleted and more details should be given in the introduction about effect of polysaccharides on diabetes.

Materials and methods:

Please check for spaces between some words that are joined together.

Line 68: no need for this subheading as it doesn’t show antidiabetic activity.

Line 71: ‘…water was boiled for 5-7 days’, this is confusing, please clarify.

Line 72: how long was the adaptation period? And clarify the random method of choosing your control.

Line 80-81: was there any basis of choosing the ACP dose (100, 400 mg), if so please state that.

Line 83: how long was the fasting period before measuring FBG?

Line 84: rat food wasn’t mentioned before, please be consistent.

Line 89: please clarify the period of fasting (food deprivation) 12 or 24 hours.

Line 91: why visceral index in this section (biochemical analysis). It should be separated.

Results:

Nothing in the results was mentioned about the extracted polysaccharide or its composition or fibre content. Nothing was mentioned about the diet and its composition of polysaccharides, fibre, energy etc. were all group isocaloric? Was the total amount of feed measured? , if so we need to know if all group had similar amount of food as less energy intake means less organ size and less body weight.

The results in general need some organisation. The authors can group the results into sections as: body weight and feed intake, blood analysis, body tissues.

Discussion:

This is a very general discussion with no critical aspects related to the results. The authors should explain their results critically in this section not as mentioned in the results section.

 The manuscript needs major modifications.

Regards

Author Response

Response to Reviewer 1 Comments

Thank you for your comments concerning our manuscript entitled “Anti-diabetic Activity of Polysaccharides from Auricularia cornea var. Li” (Foods-1714878). Those comments are all valuable and very helpful for revising and improving our paper, as well as the important guiding significance to our researches. We have studied comments carefully and have made correction which we hope meet with approval. The corrections in the paper and the responds to the reviewer’s comments are as follows:

Introduction

Point 1: The last paragraph in the introduction should be deleted and more details should be given in the introduction about effect of polysaccharides on diabetes.

Response 1: Thank you so much for the time and effort to review our manuscript. Thank you very much for the valuable comments. The last paragraph has deleted.Our manuscript have given in the introduction about effect of polysaccharides on diabetes. Please see line 51-65.

Materials and methods

Point 2: Please check for spaces between some words that are joined together.

Response 2: Thank you very much for the valuable comments. I apologize for the error. We have made correction and marked yellow.  

Point 3: Line 68: no need for this subheading as it doesn’t show antidiabetic activity.

Response 3: Thank you very much for the valuable comments. We have deleted this subheading.

Point 4:Line 71: ‘…water was boiled for 5-7 days’, this is confusing, please clarify.

Response 4: Thank you very much for the valuable comments. I apologize for the unclear meaning of the sentence. We have made correction and marked yellow. “Afterward, all animals were raised under the following conditions: temperature of 24 â—¦C, and light/dark cycle of 12 h/ 12 h. They were allowed to drink water and eat food freely. ” Please see line 97-99.

Point 5:Line 72: how long was the adaptation period? And clarify the random method of choosing your control.

Response 5: Thank you very much for the valuable comments. (1) Adapt to seven days. (2) SPSS 23.0 experimental design Random grouping method:â‘  Numbering and defining variables: number 50 mice from 1 to 50. In the variable view in the lower left corner of SPSS, you can change the name, type, width, decimal places, etc. of the variable.â‘¡Set random seed (when the random seed is fixed, a fixed random number will be generated): click "conversion" → "random number generator"; In the pop-up "random number generator" window, select "set starting point" - "fixed value", enter the fixed value, fill in the default value here, and then click "OK".â‘¢ Generate random number: click "convert" → "calculate variable"; In the pop-up "calculate variable" window, enter the name of "target variable", and enter "random number" here; Find the "RV. Uniform" function in the function group "random number", double-click the function or click the upward arrow on the right to delete, and the selected function will be selected into the "number expression" list - "RV. Uniform (?,?), In (?,?) Enter the interval generated by random numbers. This experiment is divided into five groups, so there are four intervals. Enter "0, 4" here, and then click "OK".â‘£ Use visual binning to group: click "convert" → "visual binning"; In the pop-up "visual box sorting" window, select "random number" into the "variables to box sorting" list, and click "continue". In the pop-up window, enter the name in "boxed variable", enter "grouping" here, and click "generate split point". Click "equal points" in the "split points" window, and then click "OK" to return to the "split points" window. As shown in Table 1, animals were randomly divided into groups.

Table 1. Animals were randomly divided into groups

Number

Random number

Five groups

 Groups

1

0.56

2

Model

2

1.73

3

Positive control

3

2.45

4

ACP low dose

4

1.16

3

Positive control

5

0.62

2

Model

6

2.8

4

ACP low dose

7

1.39

3

Positive control

8

1.78

3

Positive control

9

0.21

1

Normal

10

0.41

1

Normal

11

0.56

2

Model

12

0.17

1

Normal

13

2.49

4

ACP low dose

14

0.61

2

Model

15

2.86

5

ACP high dose

16

3.71

5

ACP high dose

17

2.31

4

ACP low dose

18

1.05

3

Positive control

19

2.9

5

ACP high dose

20

0.15

1

Normal

21

0.4

1

Normal

22

2.93

5

ACP high dose

23

0.9

2

Model

24

0.89

2

Model

25

2.44

4

ACP low dose

26

3.82

5

ACP high dose

27

0.95

2

Model

28

1.35

3

Positive control

29

0.21

1

Normal

30

2.95

5

ACP high dose

31

0.48

1

Normal

32

0.24

1

Normal

33

0.75

2

Model

34

0.4

1

Normal

35

2.7

4

ACP low dose

36

2.77

4

ACP low dose

37

3.47

5

ACP high dose

38

1.4

3

Positive control

39

1.99

3

Positive control

40

2.81

4

ACP low dose

41

0.82

2

Model

42

3.09

5

ACP high dose

43

0.27

1

Normal

44

2.11

4

ACP low dose

45

2.65

4

ACP low dose

46

3.74

5

ACP high dose

47

1.26

3

Positive control

48

2.09

3

Positive control

49

3.06

5

ACP high dose

50

0.51

2

Model

Point 6:Line 80-81: was there any basis of choosing the ACP dose (100, 400 mg), if so please state that.

Response 6: Thank you very much for the valuable comments. The dosage of ACP was selected according to reference [25].We have made correction and marked yellow. Please see line 114.

The reference follow:

[25]Risheng, et al.Physicochemical properties of polysaccharides separated from Camellia oleifera Abel seed cake and its hypoglycemic activity on streptozotocin-induced diabetic mice. International Journal of Biological Macromolecules (2018).

Point 7:Line 83: how long was the fasting period before measuring FBG?

Response 7: Thank you very much for the valuable comments. Fasting blood glucose was measured every 7 days. The fasting time is 12 hours.

Point 8:Line 84: rat food wasn’t mentioned before, please be consistent.

Response 8: Thank you very much for the valuable comments. I apologize for the error. We have made correction and marked yellow. Please see line 117.

Point 9:Line 89: please clarify the period of fasting (food deprivation) 12 or 24 hours.

Response 9: Thank you very much for the valuable comments. I apologize for the error. We have made correction and marked yellow. The period of fasting (food deprivation) 12 hours. Please see line 118, 128.

Point 10:Line 91: why visceral index in this section (biochemical analysis). It should be separated.

Response 10: Thank you very much for the valuable comments.We quite agree with the reviewer. I apologize for the error. We have made correction and marked yellow. Please see line 120-125.

Results

Point 11:Nothing in the results was mentioned about the extracted polysaccharide or its composition or fibre content.

Response 11: Thank you very much for the valuable comments.We quite agree with the reviewer. I apologize for the error. We have made correction and marked yellow. “The extraction rate and purity of ACP was 28.18±1.41% and 86.92±2.80%. The monosaccharides of ACP were 32.41% mannitol, 6.96% glucuronic acid, 0.32% rhamnose, 42.35% glucose, 0.77% galactose, 16.83% xylose and 0.36% fucose, without galacturonic acid and arabinose.” Please see line 148-153.

Point 12: Nothing was mentioned about the diet and its composition of polysaccharides, fibre, energy etc. were all group isocaloric? Was the total amount of feed measured? , if so we need to know if all group had similar amount of food as less energy intake means less organ size and less body weight.

Response 12: Thank you very much for the valuable comments.We quite agree with reviewer. I apologize for the error. We have made correction and marked yellow. “ The basal diet:3.40Kcal/g, protein 23.07%, fat 11.85%, carbohydrate 65.08%, Beijing Keao Xieli Feed Co.,Ltd; The high-fat and sugar diet:4.70Kcal/g, protein 19.80%, fat 35.20%, carbohydrate 45.00%, Beijing Botai Hongda Biotechnology Co., Ltd., batch No.: HD001.” Please see line 100-103.

Point 13:The results in general need some organisation. The authors can group the results into sections as: body weight and feed intake, blood analysis, body tissues.

Response 13:Thank you very much for the valuable comments.We quite agree with the reviewer. We have made correction and marked yellow. Please see “3.2 Body weight and feed intake (line 155) ,3.3 Blood analysis(line 199), 3.4 Body tissues(line 291) ”.

Discussion

Point 14:This is a very general discussion with no critical aspects related to the results. The authors should explain their results critically in this section not as mentioned in the results section.

Response 14:Thank you very much for the valuable comments.We quite agree with the reviewer. I apologize for the error. We have made correction and marked yellow. Please see “ 4 Discussion”.

We tried our best to improve the manuscript. We appreciate for Editors/Reviewers’ warm work earnestly, and hope that the correction will meet with approval.

Once again, thank you very much for your comments and suggestions, and hope you happy every day.

Reviewer 2 Report

Dear Authors,

You paper is fine, not innovative, but expanding the knowledge. Other than minor revs there are some important features to fix, like giving more, more international, and fresher references related to any section of your paper (introduction, materials and methods and discussion of results). Additionally, the abstract lacks a rationale. The last two sections (discussion and conclusion) are poor and should reinforced, for example highlighting the pros and giving some cons, and also give perspectives and applications.

Abstract:

The rationale of the work is missing, please include it.

Which is the reason to introduce new sources of anti-diabetic medications (as you have reported in the introduction)?

Line 14: Polysaccharides…correct typos

Please introduce your research pipeline in the abstract

Please give some pos an cons of your research

Please give some perspectives and applications of your research

Keywords:

Please avoid to use those words already present in the title

Introduction

Line 36: anti-tumor (please avoid to use such generic claim for such important disease), It would be better if you can specify the anti-tumor characteristics that has been found in that research. For example, anti-tumor activity versus Caco-2 cells or anti-tumor on experimental mice…

Line 42: Auricularia cornea var. Li …I am not quite sure that the style of the name is fine. Please check. “var.” should not goes in italic and also “Li”, sounds like a taxonomist, not a latin name.

Line 44: Auricularia Heimer…see the comment above

Lines 44-47 Auricularia cornea….quality

This section is a bit confused and should be revised. Additionally, is the specimen of this fungus deposited?

Materials and Methods…the sections of this chapter are a bit hard to follow, could you please try an easier way to number subparagrpahs?

2.2 Extraction of ACP  check for typos

2.3.1 Animal experimental design ….have you done something to reduce the animal testing in accordance within the concept of 3R or in accordance with EPA, FDA and EFSA or EARA (Directive 2010/63/EU) (https://www.eara.eu/animal-research-law?lang=it)  ? It would be nice if you can write about it. Your work could become more appealing for EU and US readers.

Line 91: Determination of visceral indexes…Please report a reference

Line 98: (2) Determination of hypoglycemic activity…please report the commercial kits name and their commercial supplier

2.4 Preparation of tissue…please report the commercial suppliers of the chemicals and th apparatus, and also the concentration of the chemicals used. A reference is appreciated

Line 109: Which was the program employed for?

Line 113: All experiments were run in triplicate (n=3) What do you mean, three biological replcias or three technical replicas, please explain.

Results:

Table 1..check for errors and typos.

Line 141: 3.2 Effect of ACPon blood glucose (GLU) in mice…Please correct

Line 142: had a significant numerical change…sounds strange, please use other words

Figure 1: the numbers 1,2,3,4,5 are to be deleted from the plots.

Discussion and Conclusion are a bit poor and should be implemented.

Also, alike in the introduction the reference list is short and should be implemented.

Also and very important the authors should cite not just Chinese papers.

Author Response

Response to Reviewer 2 Comments

Thank you for your comments concerning our manuscript entitled “Anti-diabetic Activity of Polysaccharides from Auricularia cornea var. Li” (Foods-1714878).  Those comments are all valuable and very helpful for revising and improving our paper, as well as the important guiding significance to our researches. We have studied comments carefully and have made correction which we hope meet with approval. The corrections in the paper and the responds to the reviewer’s comments are as follows:

Dear Authors,

Point 1: You paper is fine, not innovative, but expanding the knowledge. Other than minor revs there are some important features to fix, like giving more, more international, and fresher references related to any section of your paper (introduction, materials and methods and discussion of results). Additionally, the abstract lacks a rationale. The last two sections (discussion and conclusion) are poor and should reinforced, for example highlighting the pros and giving some cons, and also give perspectives and applications.

Response 1:Thank you so much for the time and effort to review our manuscript. Thank you very much for the valuable comments. We quite agree with the reviewer. We have made correction and marked yellow. Please see abstract, inntroduction, materials and methods, discussion and conclusion, reference.

Point 2:Abstract:

The rationale of the work is missing, please include it.

Which is the reason to introduce new sources of anti-diabetic medications (as you have reported in the introduction)?

Line 14: Polysaccharides…correct typos

Please introduce your research pipeline in the abstract

Please give some pos an cons of your research

Please give some perspectives and applications of your research

Response 2:Thank you so much for the time and effort to review our manuscript. Thank you very much for the valuable comments. We quite agree with the reviewer. I apologize for the typos error. The abstract have made correction and marked yellow. Please see “Auricularia cornea var. Li. polysaccharide (ACP) has many important biological activities and has potential application value in food engineering, pharmaceutical science and health care. The results were as follows: the extraction rate of ACP was 28.18 ± 1.41%, and the purity of ACP was 86.92 ± 2.80%; ACP contains mannitol 32.41%, glucuronic acid 6.96%, rhamnose 0.32%, glucose 42.35%, galactose 0.77%, xylose 16.83%, fucose 0.36%, without galacturonic acid and arabinose. In addition, the results of animal test of Diabetes mellitus II (DM II) with ACP showed that the total cholesterol (TC), triglyceride (TG), low density lipoprotein cholesterol (LDL-C) and fasting blood glucose and water in the serum of mice with ACP were significantly lower than those in the model group; The serum SOD, hepatic glycogen and insulin of mice added with ACP were significantly higher than those in the model group. More importantly, ACP had no significant adverse effects on organ index and liver and kidney tissue morphology in mice. These results suggest that ACP can be used as a potential functional food component for the prevention or treatment of diabetes.” (line 16-29)

Point 3: Keywords:Please avoid to use those words already present in the title

Response 3: Thank you very much for the valuable comments. The keywords and their use in the manuscript were checked. However, keywords were preserved due to their essential value for the discussion of the manuscript. Their absence would remove essential content for the comprehension of our review. We agree with the reviewer about the necessity to discuss in detail the studies composing our paper.

Point 4:Introduction

Line 36: anti-tumor (please avoid to use such generic claim for such important disease), It would be better if you can specify the anti-tumor characteristics that has been found in that research. For example, anti-tumor activity versus Caco-2 cells or anti-tumor on experimental mice…

Response 4:Thank you very much for the valuable comments. We quite agree with the reviewer. We have made correction and marked yellow. “antitumor(For example,Polysaccharide isolated from Calocybe indica var. APK2 also showed effective antitumor activities [12]. The cytotoxic effect of this polysaccharide was evaluated in vitro on HeLa cells (Human cervical cancer) and normal cells NIH3T3 (Murine fifibroblat) by the MTT assay.A water soluble polysaccharide (TOP-2, Mw :63 kDa), isolated from the basidiomycetous fungus Trametes orientalis [13]. TOP-2 showed antitumor activities in Lewis lung carcinoma (LLC) tumor-bearing mice. )” Please see line 45-51.

Point 5: Line 42: Auricularia cornea var. Li …I am not quite sure that the style of the name is fine. Please check. “var.” should not goes in italic and also “Li”, sounds like a taxonomist, not a latin name.

Response 5: Thank you very much for the valuable comments. I apologize for the error. We have made correction and marked yellow. “Auricularia cornea var. Li.” Please see line 66-73.

Point 6: Line 44: Auricularia Heimer…see the comment above.

Response 6: Thank you very much for the valuable comments. I apologize for the error. We have made correction and marked yellow. “Auricularia heimer” Please see line 66-73.

Point 7: Lines 44-47 Auricularia cornea….quality. This section is a bit confused and should be revised. Additionally, is the specimen of this fungus deposited?

Response 7: Thank you very much for the valuable comments. I apologize for the error. We have made correction and marked yellow. The fungus has been preserved in the fungus center of Jilin Agricultural University.

Point 8: Materials and Methods…the sections of this chapter are a bit hard to follow, could you please try an easier way to number subparagrpahs?

Response 8: Thank you very much for the valuable comments. I apologize for the error. We have made correction and marked yellow. Please see 2. Materials and Methods.

Point 9: 2.2 Extraction of ACP check for typos

Response 9: Thank you very much for the valuable comments. I apologize for the typos error. We have made correction and marked yellow. Please see line 86-88.

Point 10: 2.3.1 Animal experimental design ….have you done something to reduce the animal testing in accordance within the concept of 3R or in accordance with EPA, FDA and EFSA or EARA (Directive 2010/63/EU) (https://www.eara.eu/animal-research-law?lang=it)  ? It would be nice if you can write about it. Your work could become more appealing for EU and US readers.

Response 10: Thank you very much for the valuable comments. We quite agree with you. We have made correction and marked yellow. “The animal protection directive aims to protect animals in scientific researc. Furthermore, it announces the implementation of the 3Rs principle (replacement, reduction, refinement). Animal model of this experiment is in line with the 3Rs policy.” Please see line 107-110.

Point 11: Line 91: Determination of visceral indexes…Please report a reference

Response 11: Thank you very much for the valuable comments. References have been inserted.We have made correction and marked yellow. Please see line 123 [24]

Point 12: Line 98: (2) Determination of hypoglycemic activity…please report the commercial kits name and their commercial supplier.

Response 12: Thank you very much for the valuable comments.We quite agree with the reviewer.The commercial kits name and their commercial supplier see to 2.1 Materials and reagents (The commercial kits of Total cholesterol (T-CHO) kit, Triglyceride (TG) kit, High density lipoprotein cholesterol (HDL-C), Low density lipoprotein cholesterol (LDL-C), Superoxide dismutase (SOD), Ins ELISA Kit (INS) and Glycogen test box were obtained from Nanjing Biotechnology Co. Ltd. (Nanjing, China).

Point 13: 2.4 Preparation of tissue…please report the commercial suppliers of the chemicals and th apparatus, and also the concentration of the chemicals used. A reference is appreciated

Response 13: Thank you very much for the valuable comments. We have made correction and marked yellow. “10% neutral formalin; Gradient ethanol elution: 70%, 80%, 95%, 100%; Embedding machine (KH-BL, Hubei Xiaogan kuohai Medical Technology Co., Ltd.); Slicer (RM2125RT paraffin slicer, Leica, Germany); Microscope (BX53 microscope, Olympus, Japan).” Please see line 133-141.

Point 14: Line 109: Which was the program employed for?

Response 14: Thank you very much for the valuable comments. Statistical analysis has been modified. We have made correction and marked yellow. “All data were expressed as the mean ± standard deviation (SD) from 6 independent experiments and compared using one-way analysis of variance (ANOVA) by employing SPSS 23.0 software and Dunnett’s test. In addition, P-value <0.05 was considered significant.”  Please see line 142-146.

Point 15:Line 113: All experiments were run in triplicate (n=3) What do you mean, three biological replcias or three technical replicas, please explain.

Response 15: Thank you very much for the valuable comments. I apologize for the error. Statistical analysis has been modified. We have made correction and marked yellow. “All data were expressed as the mean ± standard deviation (SD) from 6 independent experiments and compared using one-way analysis of variance (ANOVA) by employing SPSS 12.0 software and Dunnett’s test. In addition, P-value <0.05 was considered significant.” Six biological replcias. Please see line 142-146.

Point 16: Results:Table 1..check for errors and typos.

Response 16: Thank you very much for the valuable comments. I apologize for the typos error. We have made correction and marked yellow. Please see Table 1.

Point 17: Line 141: 3.2 Effect of ACPon blood glucose (GLU) in mice…Please correct

Response 17: Thank you very much for the valuable comments. I apologize for the error. We have made correction and marked yellow. “3.2 Effect of ACP on blood glucose (GLU) in mice” Please see line 200.

Point 18: Line 142: had a significant numerical change…sounds strange, please use other words

Response 18: Thank you very much for the valuable comments. We have made correction and marked yellow. “As shown in Table 3, the blood glucose level of DM II mice after feeding with ACP had an obvious downward trend.” Please see line 201-202.

Point 19: Figure 1: the numbers 1,2,3,4,5 are to be deleted from the plots.

Response 19: Thank you very much for the valuable comments. We have made correction and marked yellow. The numbers 1,2,3,4,5 have deleted from the plots. Please see Figure 1.

Point 20: Discussion and Conclusion are a bit poor and should be implemented.

Response 20: Thank you very much for the valuable comments. We have made correction and marked yellow. Please see line 391-470.

Point 21:Also, alike in the introduction the reference list is short and should be implemented.Also and very important the authors should cite not just Chinese papers.

Response 21: Thank you so much for the time and effort to review our manuscript. The references and their use in the manuscript were checked. However, some references were preserved due to their essential value for the discussion of the manuscript. Their absence would remove essential content for the comprehension of our review. We agree with the reviewer about the necessity to discuss in detail the studies composing our paper. Please see reference. The referemce have updated.

We have made all modifications by the PDF of ou provided. Please see the manuscript. We tried our best to improve the manuscript. We appreciate for Editors/Reviewers’ warm work earnestly, and hope that the correction will meet with approval.

Once again, thank you very much for your comments and suggestions, and hope you happy every day.

Round 2

Reviewer 2 Report

Dear authors, thanks for the revisins made.